# DNA Methyltransferase 1 (DNMT1) Acts on Neurodegeneration by Modulating Proteostasis-Relevant Intracellular Processes

**DOI:** 10.3390/ijms21155420

**Published:** 2020-07-30

**Authors:** Cathrin Bayer, Georg Pitschelatow, Nina Hannemann, Jenice Linde, Julia Reichard, Daniel Pensold, Geraldine Zimmer-Bensch

**Affiliations:** Division of Functional Epigenetics in the Animal Model, Institute for Biology II, RWTH Aachen University, Worringerweg 3, 52074 Aachen, Germany; bayer@bio2.rwth-aachen.de (C.B.); georg.pitschelatow@rwth-aachen.de (G.P.); nina.hannemann@rwth-aachen.de (N.H.); jenice.linde@rwth-aachen.de (J.L.); reichard@bio2.rwth-aachen.de (J.R.); pensold@bio2.rwth-aachen.de (D.P.)

**Keywords:** DNMT1, autophagy, degradation, aggresome, HTT, retrograde trafficking, Huntington’s disease

## Abstract

The limited regenerative capacity of neurons requires a tightly orchestrated cell death and survival regulation in the context of longevity, as well as age-associated and neurodegenerative diseases. Subordinate to genetic networks, epigenetic mechanisms, such as DNA methylation and histone modifications, are involved in the regulation of neuronal functionality and emerge as key contributors to the pathophysiology of neurodegenerative diseases. DNA methylation, a dynamic and reversible process, is executed by DNA methyltransferases (DNMTs). DNMT1 was previously shown to act on neuronal survival in the aged brain, whereby a DNMT1-dependent modulation of processes relevant for protein degradation was proposed as an underlying mechanism. Properly operating proteostasis networks are a mandatory prerequisite for the functionality and long-term survival of neurons. Malfunctioning proteostasis is found, inter alia, in neurodegenerative contexts. Here, we investigated whether DNMT1 affects critical aspects of the proteostasis network by a combination of expression studies, live cell imaging, and protein biochemical analyses. We found that DNMT1 negatively impacts retrograde trafficking and autophagy, with both being involved in the clearance of aggregation-prone proteins by the aggresome–autophagy pathway. In line with this, we found that the transport of GFP-labeled mutant huntingtin (HTT) to perinuclear regions, proposed to be cytoprotective, also depends on DNMT1. Depletion of *Dnmt1* accelerated perinuclear HTT aggregation and improved the survival of cells transfected with mutant HTT. This suggests that mutant HTT-induced cytotoxicity is at least in part mediated by DNMT1-dependent modulation of degradative pathways.

## 1. Introduction

The limited regenerative capacity of neurons requires tight orchestration of neuronal functionality and survival [1]. Epigenetic mechanisms of transcriptional control, such as DNA methylation, catalyzed by DNMTs, as well as histone modifications, contribute to cell death regulation during development, aging, and in disease [2,3]. DNMT1 is one of the main DNMTs expressed in the developing and adult brain. In addition to its function in dividing progenitors, DNMT1 catalyzes promoter as well as gene body cytosine methylation in post-mitotic and adult neurons [4,5,6,7]. DNMT1 has already been reported to promote the survival of a variety of maturing neurons, including excitatory neocortical and hippocampal neurons [8], photoreceptor cells, and other types of neurons in the postnatal retina [9]. Moreover, we found that DNMT1 is crucial for the survival of developing γ-aminobutyric acid (GABA)-ergic inhibitory interneurons [10].

In contrast to its survival-promoting function in developing neural cells, we found that DNMT1 negatively influences cortical interneuron longevity in the aged brain [11]. However, no cell death or survival-related genes have been revealed to be targeted by DNMT1 function as an underlying cause. In turn, a significant enrichment of genes associated with proteostasis pathways were found to be suppressed by DNMT1, through which the effect on interneuron longevity could be mediated [10,11,12].

The proteostasis network ultimately impacts on long-term health of neurons. Its decline causing ineffective protein degradation can lead to neuronal death and has emerged as being implicated in age- and disease-related neurodegeneration [13]. Lysosomal degradation is of great importance for removing defective proteins or protein aggregates delivered by autophagy- or endocytosis-triggered endosomal pathways [14,15,16]. Hence, it is not surprising that lysosomal dysfunction is associated with neurodegenerative pathologies [17,18]. Similarly, autophagy, as the major intracellular machinery for degrading aggregated proteins and damaged organelles, is implicated in the disease pathology of many neurodegenerative disorders [19,20].

Numerous neurodegenerative diseases are characterized by aggregation-prone proteins, which are suggested to cause or to contribute to the disease pathophysiology. For example, Huntington’s disease (HD) is caused by the expansion of CAG base triplet repeats coding for glutamine (Q) in exon 1 of the huntingtin gene. These polyQ repeats lead to misfolding of the mutant huntingtin (HTT) protein, which are highly prone to aggregate [21,22,23,24,25].

It is proposed that misfolded proteins are actively transported to a cytoplasmic juxtanuclear structure, called an “aggresome”, when the chaperone refolding system and the ubiquitin–proteasome degradation pathway are overwhelmed by the production of such misfolded proteins [26]. Aggresome formation is considered a cytoprotective response serving to sequester potentially toxic misfolded proteins into pericentriolar inclusion bodies and facilitate their clearance by autophagy [27,28,29,30,31,32,33,34]. The aggresome–autophagy pathway, proposed to concertedly eliminate such aggregation-prone proteins such as mutant HTT [28], is coupled to the retrograde microtubule-based transport to recruit both the aggregated proteins as well as molecular determinants of autophagic vacuole formation and lysosomes to pericentriolar cytoplasmic inclusion bodies [32,33,34]. In support of this, microtubule-disrupting agents prevent aggresome formation, and result in elevated polyglutamine toxicity [30,35].

In the context of HD, it was shown that DNMTs mediate the mutant HTT-induced cytotoxic effects in cortical neurons, whereas the underlying mechanism is unknown. We have evidence that DNMT1 represses the expression of genes related to autophagy, perinuclear region, and intracellular microtubule trafficking events, such as endo-lysosomal transport, being crucial for degradative proteostasis networks [12]. Indeed, we found accelerated retrograde endosomal transport and trafficking of endocytosed cargo fated for lysosomal degradation upon *Dnmt1* depletion [11]. As retrograde trafficking is central to the aggresome–autophagy pathway, and as neurons expressing polyglutamine-expanded *Htt* show improved survival when they form aggresomes [36], we here aimed to investigate whether DNMT1 is implicated in mutant HTT-induced cytotoxicity by acting on retrograde transport, aggresome formation, and autophagy.

## 2. Results

### 2.1. DNMT1 Modulated Retrograde Trafficking

In previous studies, we identified that DNMT1 transcriptionally controls a variety of endosomal, lysosomal, and retrograde trafficking-associated genes in their expression, with most of them being repressed by DNMT1 in cortical interneurons [11,12] (Figure 1a and Appendix A). Functionally, we confirmed that DNMT1 slows down the velocity of retrograde transportation of endolysosomal compartments in cerebellar granule (CB) cells [11], which we frequently use as a neuronal cell culture model [11,12]. Here, we verified whether retrograde transport is likewise modulated by DNMT1 in neuroblastoma (N2a) cells, in order to assess whether the regulation of retrograde transport is a more general, cellular sub-type independent function of DNMT1. To this end, we transfected N2a cells with cluster of differentiation 63-green fluorescent protein (*Cd63-GFP*) and lysosomal-associated membrane protein 1 (*Lamp1*)*-mCherry*-encoding plasmids in order to label endosomes and lysosomes [37], with both being involved in autophagy and endocytosis-dependent degradation [38]. We then analyzed the velocity of anterogradely and retrogradely moving CD63- and LAMP1-containing compartments upon *Dnmt1* siRNA knockdown by using live cell imaging (Figure 1b–f and Appendix A). Compared to control siRNA treatment, *Dnmt1* depletion resulted in significantly elevated velocities of retrogradely moving CD63- and LAMP1-positive particles, while the anterograde transportation was not affected (Figure 1d,f). In line with this, the overall distances of retrogradely but not anterogradely transported CD63- and LAMP1-positive compartments were found to be significantly increased after *Dnmt1* knockdown (Appendix A). Together, this suggests that DNMT1 selectively influences retrograde trafficking across different neuronal model systems.

### 2.2. DNMT1-Mediated Retrograde Trafficking to Perinuclear Regions Influenced Mutant HTT-Induced Cytotoxicity

Degradative proteostasis networks rely on retrograde trafficking in many regards. Apart from trafficking of endosomes to degradative lysosomes or secretory compartments [39], retrograde transportation is crucial for autophagic clearance [40]. Indeed, defects in the retrograde transport of autophagosomes are involved in disease-related neurodegeneration [40,41]. Moreover, the aggresome–autophagy pathway relies on microtubule-based retrograde trafficking [26]. When the ubiquitin proteasome system is overwhelmed with aggregation-prone proteins, the aggresome–autophagy pathway sequesters misfolded proteins into pericentriolar inclusion bodies called “aggresomes” and facilitates their clearance [33,34]. Aggresome formation is suggested to improve neuronal survival upon expression of polyglutamine-repeat containing proteins such as mutant HTT, which escape proteasome-dependent degradation [42].

In line with this, we observed numerous N2a and CB cells forming GFP aggregates in the perinuclear region upon transfection with a mutant *Htt-GFP* containing plasmid and inhibition of proteasomal degradation with MG-132. The utilized plasmid encodes for mutant HTT-GFP, leading to the expression of the GFP-labeled exon 1 of the mutated Huntingtin protein with a total of 103 glutamine repeats. While HTT-GFP was initially distributed equally in the cytosol at the beginning of the live cell imaging, a localized and intense GFP spot was visible in numerous cells at 6 h of live cell imaging (Figure 2a,b and Appendix A), indicative of aggresome formation. The perinuclear localization of the HTT-GFP aggregate can be observed clearly for CB cells (Figure 2d and Appendix A). In support of a survival promoting function of aggresome formation, we found improved survival rates for N2a cells that had formed visible HTT-GFP aggregates when compared to the cells with no apparent aggregation of the mutant HTT-GFP protein (Appendix A). Similar observations in matters of improved survival upon aggregate formation were made for CB cells (Appendix A).

Interestingly, DNMTs were reported to be involved in mediating the cytotoxic effects of mutant HTT, whereas the underlying mechanism is unknown. As we found that DNMT1 restricts effective intracellular retrograde trafficking and represses genes relevant for the formation of the perinuclear region and microtubule-based processes (Appendix A), we asked whether DNMT1 could interfere with HTT-GFP aggregation in the perinuclear region and through this promote mutant HTT-induced cytotoxicity.

To this end, we used live cell imaging to monitor the survival of mutant HTT-GFP-expressing N2a and CB cells, in which proteasomal degradation was blocked after transfection with *Dnmt1* or control siRNA (Figure 3; Appendix A). While mutant HTT protein expression resulted in a severe reduction of cell survival over a time period of 12 h for both N2a and CB cells, the mutant HTT-induced cytotoxicity was significantly decreased in both cell lines upon knockdown of *Dnmt1*. *Dnmt1* depletion caused improved survival rates, similar to what was reported by Pan et al. [43] (Figure 3).

Next, we aimed to find out whether the implication of DNMT1 in mediating the mutant HTT-induced cytotoxicity depends on the DNMT1-dependent modulation of retrograde trafficking. To this end, we measured the velocity of HTT-GFP aggregate formation. For this, we focused on CB cells. Due to their larger soma size, the dynamics and the temporal course of perinuclear HTT aggregation can be clearly identified by live cell imaging, in contrast to aggregation in the smaller-sized and roundish N2a cells. In line with our findings that *Dnmt1* siRNA application increased the velocity of retrograde transport [11], we found that the formation of HTT-GFP aggregates was significantly faster upon siRNA-mediated depletion of *Dnmt1* in CB cells (Figure 4). This indicates that DNMT1 is involved in mutant HTT-induced cytotoxicity via modulation of aggresome formation by acting on retrograde transportation.

### 2.3. DNMT1 Is Relevant for Proper Autophagy

The accumulation of aggregated proteins is suggested to induce an autophagic response to concertedly eliminate aggregation-prone proteins. To this end, molecular determinants of autophagic vacuole formation and lysosomes have been reported to be recruited to the pericentriolar cytoplasmic inclusion bodies, which relies on microtubule-dependent retrograde trafficking and which we found to be decelerated by DNMT1 [28].

Interestingly, we found the autophagy-related genes to be repressed by DNMT1 as well, as depicted in Figure 5a. Hence, we next examined whether DNMT1 function modulates autophagy. Autophagy starts with the formation of a phagophore, a double membrane that encloses and isolates the cytoplasmic components, which leads to the generation of autophagosomes. Autophagosomes fuse with multivesicular bodies (MVB)/late endosomes, which matured from early endosomes, generating an amphisome. The amphisomes then get delivered to lysosomes, fusing to autolysosomes for degradation of the cargo [38] (Figure 5b). The formation of autophagic vacuoles or autophagosomes is accompanied by the conjugation of the cytosolic microtubule-associated protein 1 light chain 3B-I (LC3B-I) with phosphatidylethanolamine, which results in the formation of the membrane-associated LC3B-II [44]. We monitored alterations in LC3B-II levels using Western blot in N2a cells, which were either treated with control or *Dnmt1* siRNA (Figure 5c). Autophagy was triggered in these cells by incubation of neurons in starving medium, resulting in elevated LC3B-II levels upon *Dnmt1* siRNA treatment (Figure 5c,d). Additionally, we added bafilomycin A1, which causes the inhibition of the fusion of autophagosomes and lysosomes. This leads to an impaired degradation and an accumulation of LC3B-II-positive autophagosomes, which were likewise increased after *Dnmt1* siRNA treatment (Figure 5c,d). DNMT1 protein levels were analyzed as well, showing the clear reduction of the protein upon treatment with the respective siRNA (Figure 5c,d) and confirming the siRNA-mediated knockdown.

In sum, DNMT1 appeared to modulate autophagy. Collectively, these data suggest that DNMT1 is involved in the mutant HTT-induced cytotoxicity by acting on degradative pathways including retrograde transportation, aggresome formation, and autophagy.

## 3. Discussion

Innumerable potential mechanisms are proposed to be involved in neurodegeneration, such as defects in protein homeostasis, impaired protein degradation, alterations in gene expression and transcriptional regulation, in addition to mitochondrial dysfunction [45]. However, a prerequisite to improve early diagnosis and to develop disease-modifying therapy strategies is the detailed understanding of the interplay and hierarchies of the pathophysiological mechanisms.

Epigenomic remodeling calls increasing attention in the field of neurodegeneration, proposed to mediate the initiation and progression of neurodegenerative disorders and to potentially serve as novel targets for therapeutic interventions [46]. Epigenetic transcriptional control is known to orchestrate diverse aspects of neuronal physiology in the developing and adult brain, being essentially implicated in the regulation of neuronal survival and function [2,3,47]. Epigenomic signatures are moreover dynamically reconfigured in the aging brain, being suggested to mediate age-related alterations, such as the loss of synapses and even particular neuronal subpopulations [3,11]. Hence, it is not surprising that epigenetic dysregulation has been proposed to be implicated in events underlying neuronal dysfunction and neuronal cell death in diverse neurodegenerative diseases including Huntington’s disease (HD) [47,48,49].

HD is caused by the expansion of CAG repeat coding for glutamine (Q) in exon 1 of the huntingtin gene [50,51]. The mutated HTT protein has been reported to act on a wide range of epigenetic signatures, including histone modifications (i.e., acetylation, methylation, and ubiquitination), and DNA methylation marks [49,52]. These alterations can be associated with transcriptional dysregulation, accounting as a major pathogenic mechanism of HD and an early event in HD pathology, preceding the onset of neuronal cell death [52,53]. In line with this, DNA methylation signatures were found to be changed in HD patients and HD animal models, correlating with alterations in gene expression (reviewed in [47]). Pan et al. (2016) [43] reported that DNMTs, catalyzing DNA methylation, promote mutant HTT-induced cytotoxicity, whereas the underlying mechanism remains unknown. We found DNMT1 being implicated in the age-related neurodegeneration, underlining the importance of DNMT function and DNA methylation for neuronal longevity regulation [11]. Thereby, DNMT1 does not seem to target cell death- or survival-associated genes to modulate neuronal survival. Instead, DNMT1 influences neuronal homeostasis by regulating the expression of proteostasis-related genes, including endocytosis-, endosome-, lysosome-, and autophagy-related genes [11,12].

A decline in protein homeostasis is known to contribute to numerous neurodegenerative disorders [13]. Due to the limited regenerative capacity of neurons, clearance of defective proteins or protein aggregates by proteolytic degradation in lysosomes is of great importance for neuronal homoeostasis [54]. Lysosomal dysfunction as well as defective autophagy are implicated in neurodegenerative disorders [55,56,57,58,59,60,61]. In line with the transcriptional repression of autophagy-related genes, we here functionally verified that DNMT1 negatively regulates autophagy. Further, we found that DNMT1 acts as a brake on retrograde transportation of endo-lysosomal compartments. Apart from the transportation of endosomes to degradative lysosomes or secretory compartments [39], retrograde trafficking is crucial for autophagic clearance [40]. Defective retrograde trafficking of autophagosomes contributes to disease-related neurodegeneration [40,41]. Moreover, microtubule-based retrograde transportation is essential for the aggresome–autophagy pathway [26]. This pathway sequesters misfolded proteins into pericentriolar inclusion bodies called “aggresomes” and facilitates their clearance, suggested to compensate for defects in the ubiquitin proteasome-dependent degradation of aggregation-prone proteins [33,34]. Aggresome formation is proposed to improve neuronal survival upon expression of polyglutamine-repeat containing proteins, such as mutant HTT, which escape proteasome-dependent degradation [42]. It was shown that inclusion formation reduced the amount of mutant HTT in other areas of the cell, being associated with increased cell survival [36,62]. This is in line with our findings, showing elevated survival rates of mutant *Htt*-expressing cells upon aggregate formation. Interfering with aggresome formation, which relies on microtubule-based retrograde trafficking, by inhibiting microtubule polymerization or impairing dynein motor function, leads to decreased viability of cells expressing disease proteins [30]. Interestingly, we found that the retrograde transport and the aggregation of mutant HTT-GFP are accelerated and the HTT-induced cytotoxicity is reduced when *Dnmt1* is depleted. Together, this indicates that DNMT1 acts on aggresome formation by modulating retrograde transportation to perinuclear regions. In addition to the transport of aggregated proteins to perinuclear regions, retrograde trafficking is further suggested to recruit autophagy-relevant compartments such as autophagosomes and lysosomes [63]. Indeed, there is accumulating evidence that aggresomes are substrates for autophagy [27,28,29,30,31], even being proposed to concentrate aggregated proteins for more efficient autophagic degradation [28,64,65]. As DNMT1 affects both retrograde transportation and autophagy, DNMT1 could indirectly promote mutant HTT-induced cytotoxicity by lowering the efficiency of the aggresome–autophagy pathway.

However, we cannot exclude that DNMT1 directly interacts with mutant and/or wild-type HTT and possibly mediates changes in epigenomic and transcriptional remodeling. Indeed, HTT is assumed to be a multi-functional protein that participates in diverse cytosolic as well as nuclear processes [47]. Apart from its putative function in vesicular trafficking, HTT is localized in the nucleus and modulates transcription by binding to transcription factors. Moreover, HTT affects the function of epigenetic key players such as the repressor element 1-silencing transcription factor (REST) and the multi-subunit polycomb repressive complex 2 (PRC2) (reviewed in [47]). PCR2 is an epigenetic silencer complex catalyzing repressive H3K27 trimethylations [66]. Rising numbers of polyglutamine repeats in mutant HTT were shown to progressively increase the histone H3K27 tri-methylase activity of PRC2 in vitro [67]. This is in line with the finding of genome-wide changes in H3K27me3 signatures upon expression of mutant *Htt* in embryonic stem cells [68]. The connection between mutant HTT and PRC2 is further reinforced by changes in death-promoting proteins upon induced PRC2 deficiency in striatal medium spiny neurons (MSNs). Moreover, expression changes of transcription factors, which are normally suppressed in MSNs, were observed, accompanied by progressive and fatal neurodegeneration [69]. In addition to this, loss of PRC2 in forebrain neurons elevates the expression of genes involved in HD [69].

Although direct interaction of HTT and DNMT1 have to our best knowledge not been described thus far, mutant HTT could also indirectly interfere with DNMT1 function, as DNMT1 has been proposed to interact with PRC2 [70,71,72,73]. Moreover, certain histone modifications can prevent or promote DNA methylation [74], whereby DNMT1 function could be secondarily affected by mutant HTT-induced changes of the histone code. Non-coding RNAs (ncRNAs) likewise exert transcriptional control in part by recruiting or preventing the binding of proteins/complexes involved in setting up DNA methylation and histone modification marks [75]. Mutant HTT modulates REST function, which controls the expression of diverse long non-coding RNAs (lncRNAs) and micro RNAs (miRNAs), several of which are found dysregulated in HD [76]. Hence, DNMT1 function and DNA methylation targeting could be affected by the altered repertoire of ncRNAs. lncRNAs with critical functions in neuronal development [75] are known to regulate the binding of DNMTs to the DNA, thereby being involved in mediating site-specific methylation [77,78].

Hence, to judge the potential of epigenetic mechanisms as targets for therapeutic interventions for neurodegenerative diseases such as HD, we need to draw a conclusive picture. It is indispensable to shed light on the hierarchy of mutant HTT-induced epigenomic remodeling, and the function these epigenetic modifications have in healthy neurons. Moreover, we have to clearly dissect the primary effects from the compensatory responses of neurons when facing neurodegeneration. Here, we added up another part of the puzzle providing evidence that DNMT1-dependent transcriptional control lowers the efficiency of degradative processes, which could contribute to the polyQ-mediated cytotoxicity. In that regard, the decrease seen in *Dnmt1* expression in HD STHdhQ111 cells [79], as well as the diminished *Dnmt1* expression observed in the striatum and cortex of N171-82Q transgenic HD mice, may be interpreted as counterregulatory mechanisms of the cells to improve the efficacy of mutant HTT clearance.

## 4. Materials and Methods

### 4.1. Cell Lines and Cell Culture

Neuroblastoma (N2a) cells (ATCC: CCL-131) were cultured in Dulbecco’s modified Eagle’s medium with high glucose, GlutaMAX supplement, and pyruvate (DMEM; #31966-021, Thermo Fisher Scientific, Waltham, MA, USA), supplemented with 2% fetal bovine serum (FBS; Biowest, Nuaillé, France), 100 U/mL penicillin (Thermo Fisher Scientific, Waltham, MA, USA), and 100 µg/mL streptomycin (Thermo Fisher Scientific, Waltham, MA, USA) at 37 °C, 5% CO_2_, and 95% relative humidity. The cells were splitted after reaching a confluence of 80–90%.

Cerebellar granule (CB) cells [80] were cultured in Dulbecco’s modified Eagle’s medium with high glucose (DMEM; #41965-039, Thermo Fisher Scientific, Waltham, MA, USA), supplemented with 10% FBS (Biowest), 1% GlutaMAX, 24 mM KCl, 100 U/mL penicillin, and 100 µg/mL streptomycin incubated at 33 °C, 5% CO_2_, and 95% relative humidity. The cells were splitted after reaching a confluence of 75%.

### 4.2. Transfection with siRNA and Co-Transfection with siRNA and Plasmid DNA

Transfection of cells with siRNA was performed via lipofection using Lipofectamine 2000 or Lipofectamine 3000 (Thermo Fisher Scientific), according to the manufacturer’s protocol and as described in Zimmer et al. (2011) [81]. Mouse *Dnmt1* siRNA (30 nM; #sc-35203, Santa Cruz Biotechnology, Dallas, TX, USA) or control siRNA (15 nM; Block-iT Alexa Fluor red (#14750100) or Block-iT green (#2013) fluorescent oligo, Thermo Fisher Scientific, Waltham, MA, USA) were applied for 5 h in antibiotic- and serum-free Opti-MEM I Reduced Serum Medium (Thermo Fisher Scientific) and cells were then grown in respective cell culture media. For co-transfection we applied 30 nM *Dnmt1* siRNA (#sc-35203, Santa Cruz Biotechnology), 15 nM control siRNA (Block-iT Alexa Fluor red (#14750100) or Block-iT green ((#2013) fluorescent oligo, Thermo Fisher Scientific, Waltham, MA, USA) and plasmid DNA (pLAMP1-mCherry, 200 ng/µL; CD63-pEGFP C2, 200 ng/µL; 1× GFP-pEGFP N3-HTT, 260 ng/µL) with Lipofectamine 2000 (Thermo Fisher Scientific), as described in the protocol of the manufacturer. Antibiotic- and serum-free Opti-MEM I Reduced Serum Medium (Thermo Fisher Scientific) were used for preparation of transfection reagents and dilutions. Co-transfections were performed for 24 h prior to live cell imaging.

### 4.3. Monitoring of Endo-Lysosomal Vesicle Trafficking

To monitor the endo-lysosomal trafficking in N2a cells, 71 cells per mm² were seeded in wells of a 24-well imaging plate (#0030741021, Eppendorf, Hamburg, Germany) that was previously coated with 19 μg/mL laminin (Sigma-Aldrich, Darmstadt, Germany) and 10 μg/mL poly-L-lysine (Sigma-Aldrich) in Gey’s balanced salt solution (GBSS). The coating solution was applied and incubated for 30 min at 37 °C. Subsequently, coverslips were washed once with sterile water and let dry prior to cell seeding. Cells were cultivated in 500 µL cell culture medium for 24 h using the above-mentioned culture conditions before co-transfection with siRNAs and plasmid DNA (pLAMP1-mCherry, CD63-pEGFP C2) was performed. Vesicle trafficking was captured via live cell imaging after 24 h of co-transfection (detailed information on microscopy see below in Section 4.9).

### 4.4. Huntingtin Cytotoxicity Assay

For monitoring the mutant HTT-induced cytotoxicity, cells were seeded with densities of 55 cells per mm² for CB cells and 137 cells per mm² for N2a cells in a 24-well cell culture plate and were incubated for 24 h in 500 µL cell culture medium prior to co-transfection of siRNA and plasmid DNA (1× GFP-pEGFP N3-HTT). Then, 21 h after transfection, cells were incubated with the proteasome blocker carbobenzoxy-L-leucyl-L-leucyl-L-leucinal (1 µM, MG-132, Sigma-Aldrich). After 2.5 h, cells were additionally treated with the synthesis blocker cycloheximide (10 µg/mL, Sigma-Aldrich). Both inhibitors were diluted in phenol red-free cell culture medium (#21063-029, Thermo Fisher Scientific, Waltham, MA, USA). Cells were then monitored using live cell imaging for either 6 or 12 h.

### 4.5. Autophagy Assay

To measure changes in autophagy, we seeded N2a cells at a density of 112 cells per mm² in a 6-well cell culture plate in 2 mL culture medium, and then incubated them for 24 h prior to siRNA transfection. Afterwards, cells were incubated for 2 h at 37 °C in 1 mL normal culture medium, supplemented with 200 nM bafilomycin A1 (#sc-201550, Santa Cruz Biotechnology) or culture medium supplemented with DMSO. Subsequently, the medium was changed to either fresh culture medium with respective supplements (200 nM bafilomycin A1 or DMSO) or to Earle’s balanced salt solution (EBSS; #14155-048, Thermo Fisher Scientific, Waltham, MA, USA), supplemented with 200 mg/L MgSO_4_ and 200 mg/L CaCl_2_ with respective supplements (200 nM bafilomycin A1 or DMSO). After 2 h, the cells were washed and harvested in 1× PBS by rinsing the cells off the culture plate and centrifugation at 4 °C for 3 min at 800 rcf. The supernatant was discarded and the cell pellet was lysed in RIPA lysis buffer (50 mM Tris-HCl (pH 8.0), 150 mM NaCl, 0.5% NP-40, 0.1% SDS, 0.1% sodium deoxycholate freshly supplemented with 1 mM phenylmethylsulphonyl fluoride (PMSF; Serva, Heidelberg, Germany), and 1 µg/mL leupeptin (Serva)) for 10 min on ice. The lysate was treated twice for 30 s in an ultrasonic bath (100% power) and was later centrifuged at 4 °C for 20 min at 21,000 rcf. The supernatant was transferred into a new tube and the pellet was discarded. Samples were prepared for Western blot as described below.

### 4.6. Protein Preparation, Western Blotting, and Detection

The protein concentration of lysates was determined using the QuBit 4 fluorometer (Thermo Fisher Scientific, Waltham, MA, USA). Samples with a protein amount of 30 µg were boiled in 4× sample buffer (200 mM Tris (pH 6.8), 4% SDS, 10% β-mercaptoethanol, 40% glycerol, and 0.002% bromophenol blue) at 95 °C for 10 min, and subsequently stored at −20 °C. The prepared protein samples were separated on a ready-to-use 4–12% gradient gel (Serva) using 1× SDS-running buffer (Laemmli buffer, Serva) following the manufacturer’s instructions. Proteins were transferred to a nitrocellulose membrane using a semi-dry blot procedure in combination with the Towbin blotting kit (Serva) following the manufacturer’s instructions. Subsequently, membranes were blocked in 1× BlueBlock-reagent (Serva) for 1 h. Incubation with the following primary antibodies occurred overnight: rabbit-anti-DNMT1 (#70-201, BioAcademia, Osaka, Japan, 1:1000), mouse-anti-β-ACTIN (#sc-69879, Santa Cruz Biotechnology, 1:1000), rabbit-anti-LC3B (#2775S, Cell Signaling Technology, Cambridge, UK, 1:1000). The following secondary antibodies coupled to horseradish peroxidase (HRP) were applied for 1 h: sheep anti-mouse (#NA9310V, Sigma-Aldrich, 1:4000), donkey anti-rabbit (#NA9340V, Sigma-Aldrich, 1:4000). Membranes were washed after each incubation step three times for 5 min in Tris-buffered saline-Tween (TBS-T) buffer (25 mM Tris/HCl (pH 7.5), 137 mM NaCl, 2.7 mM KCl, 0.05% Tween-20). Chemo-luminescent detection of protein bands was performed at the blot documentation system (ChemDoc, BioRad, Feldkirchen, Germany) after applying the HRP substrate solution (Serva). Protein levels were normalized to β-actin.

### 4.7. Plasmid Information

pLAMP1-mCherry [82] was a gift from Amy Palmer (Addgene plasmid #45147; http://n2t.net/addgene:45147; RRID:Addgene_45147). CD63-pEGFP C2 was a gift from Paul Luzio (Addgene plasmid # 62964; http://n2t.net/addgene:62964; RRID:Addgene_62964). 1× GFP-pEGFP N3-HTT was a gift from Mukhran Khundadze (Institute of Human Genetics, University Hospital Jena, Germany).

### 4.8. Sequencing Data Analysis

The RNA sequencing data are published in Pensold et al. (2020) [12]. Briefly, for isolation of fluorescently labeled cells from adult *Pvalb-Cre/tdTomato/Dnmt1* WT as well as *Pvalb-Cre/tdTomato/Dnmt1 loxP^2^* mice, neurons were from whole brain hemispheres. Following the addition of DAPI, cells were sorted using an ARIA III FACS sorter (BD Biosciences, San Jose, CA, USA). RNA was isolated using the TRIzol (Invitrogen) protocol according to the manufacturer’s instructions. RNA quality was assessed by measuring the RIN (RNA integrity number) using the fragment analyzer from Advanced Analytical (Ames, IA, USA). Library preparation for RNA-Seq was performed using the TruSeq RNA Sample Prep Kit v2 (Illumina, San Diego, CA, Cat. N°RS-122-2002, USA) starting from 50 ng of total RNA. Accurate quantitation of cDNA libraries was performed by using the QuantiFluor dsDNA System (Promega, Madison, WI, USA). The size range of final cDNA libraries was determined by applying the DNA chip on the fragment analyzer (average 350 bp; Advanced Analytical). cDNA libraries were amplified and sequenced by using the cBot and HiSeq2000 from Illumina (SR; 1 × 50 bp; ≈ 30–40 million reads per sample). Sequence images were transformed with Illumina software BaseCaller to bcl files, which were demultiplexed to fastq files with CASAVA v1.8.2. Quality check was done via fastqc (v. 0.10.0, Babraham Bioinformatics, Cambridge, UK, G.B.). Read alignment was performed using STAR (v2.3.0; [83]) to the mm10 reference genome. Data were converted and sorted by samtools 0.1.19 and reads per gene were counted via htseq version 0.5.4.p3. Normalization of raw counts and differential gene expression analysis were performed using the DESeq2 R package (v 1.12.3; [84]). Genes were considered differentially expressed with a Benjamini–Hochberg adjusted *p*-value of * *p* < 0.05. Gene lists were submitted to the Database for Annotation, Visualization and Integrated Discovery (DAVID, https://david.ncifcrf.gov) for Gene Ontology enrichment analysis. Results of GO enrichment analysis were visualized in a bar diagram, including the respective Benjamini–Hochberg-corrected *p*-value and the number of genes. Heatmaps were generated using R package pheatmap (https://CRAN.R-project.org/package=pheatmap).

### 4.9. Microscopy and Data Analysis

For detection of CD63-GFP and LAMP1-mCherry proteins signals in N2a cells in the endosomal and lysosomal vesicle trafficking assay, we imaged cells using the DMi8 inverted microscope equipped with the thunder imaging platform (Leica, Wetzlar, Germany), 40× oil objective, and an incubation chamber (37 °C, 5% CO_2_). Brightfield, FITC, and TRITC channels were used to check the transfection success of cells prior to time-lapse imaging. CD63-GFP was detected by applying FITC filter settings with 12 ms exposure time at 470 nm excitation wavelength. LAMP1-mCherry was detected using TRITC filter settings with 12 ms exposure time at 550 nm excitation wavelength. Time-lapse images were acquired with a resolution of 1485 × 932 pixels in the respective channel of the fluorescent tag of the fusion protein. Transfected cells were imaged with z-stacks (five stacks, 0.494 µm z resolution at 488 nm) focused onto the neurites, covering in total 2 to 2.3 µm depending on the emission wavelength of the fluorescent tag. One imaging interval through all five stacks and respective channels was taken every 600 ms for a total time period of 90 s, resulting in 150 timepoints per cell. Instant computational clearing (Thunder Lightning, Leica) for confocal-like imaging was applied prior to data analysis. Endosomal and lysosomal vesicle movement speed, transport direction, and distances were tracked and analyzed with the ImageJ software (version 1.52p) on maximum intense projections of the z-stacks.

CB and N2a cells in the Huntingtin cytotoxicity assay were imaged with the DMi8 inverted microscope with thunder imaging platform (Leica) with 20x objective and incubation (33°C for CB cells, 37 °C for N2a cells, both 5% CO_2_). Cells were imaged for 6 h or 12 h, capturing an image every 30 min. Huntingtin-GFP was detected in the FITC channel, and cells were additionally imaged in brightfield. ImageJ (version 1.52p) was used for the analysis of cell survival and tracking of Huntingtin-GFP signal accumulation [85]. Photoshop CC (version 21.2) (Adobe Inc., San José, CA, USA) was applied for image illustration. Graphs and boxplots were generated using Microsoft Excel (version 2019) (Microsoft Corporation, Redmond, WA, USA). Significance was analyzed with two-tailed Student’s *t*-test or two-way ANOVA. Significance levels: * *p* value < 0.05; ** *p* value < 0.01; *** *p* value < 0.001. If not stated differently, experiments were repeated three times.

### 4.10. Data and Materials Availability:

RNA-sequencing data of FAC-sorted *Pvalb-Cre/tdTomato/Dnmt1* WT and *Pvalb-Cre/tdTomato/Dnmt1 loxP^2^* samples will be provided on GEO. All other data are available in the main text or the Appendix A section.

## Figures and Tables

**Figure 1 ijms-21-05420-f001:**
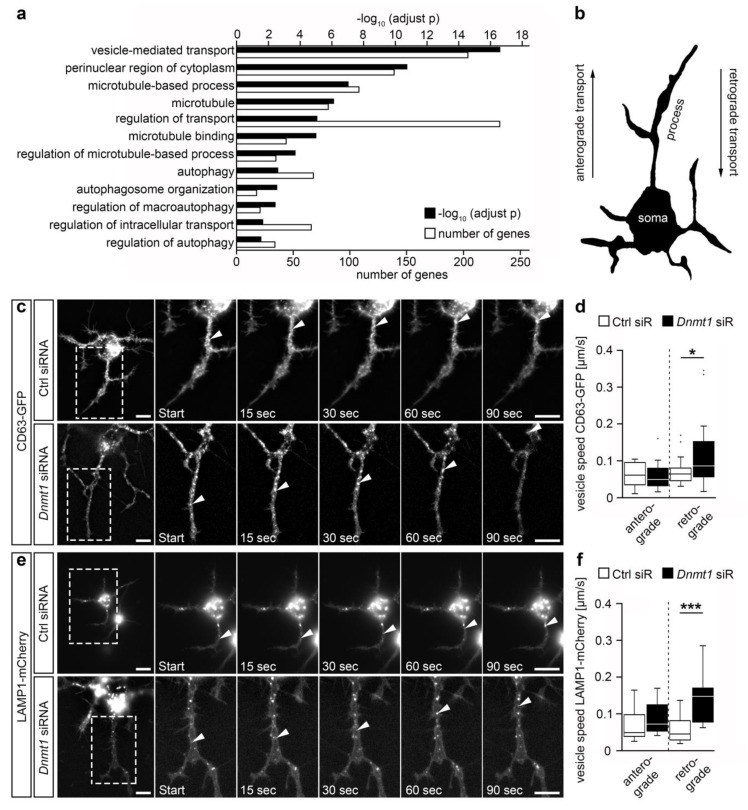
DNMT1 acts on retrograde vesicle transport. (**a**) GO terms found significantly enriched for genes that were up-regulated upon *Dnmt1* deletion in adult *Pvalb-Cre*-expressing cortical interneurons determined by RNA sequencing after fluorescence activated cell (FAC)-sorting of *Pvalb-Cre*/*tdTomato*/*Dnmt1* wild type (WT) and knockout (KO) mice (Benjamini-adjusted, *n* = 9 WT and *n* = 12 KO mice). (**b**) Schematic representation of a cell, depicting the directions of intracellular transport in the neurite-like processes. (**c**) Representative microphotographs of the tracking of cluster of differentiation 63 (CD63)-green fluorescent protein (GFP)-labeled vesicles during live cell imaging in neuroblastoma (N2a) cells transfected with *Cd63-GFP* plasmid together with control (Ctrl) siRNA (*n* = 14 cells and 34 vesicles) or *Dnmt1* siRNA (*n* = 14 cells and 38 vesicles). The velocity of the transported vesicles is quantified in (**d**) (two-tailed Student’s *t*-test, * *p* < 0.05). (**e**) Representative microphotographs of the tracking of lysosomal-associated membrane protein 1 (LAMP1)-mCherry-positive vesicles during live cell imaging in N2a cells transfected with *Lamp1-mCherry* plasmid together with Ctrl siRNA (*n* = 20 cells and 38 vesicles) or *Dnmt1* siRNA (*n* = 17 cells and 37 vesicles). The velocity of the transported vesicles is quantified in (**f**) (Two-tailed Student’s *t*-test, *** *p* < 0.001). Ctrl = control, KO = knockout, LAMP1 = lysosomal-associated membrane protein 1, N2a = neuroblastoma cells, siR = siRNA, WT = wild type. Scale bars: 10 µm in (**c**,**e**).

**Figure 2 ijms-21-05420-f002:**
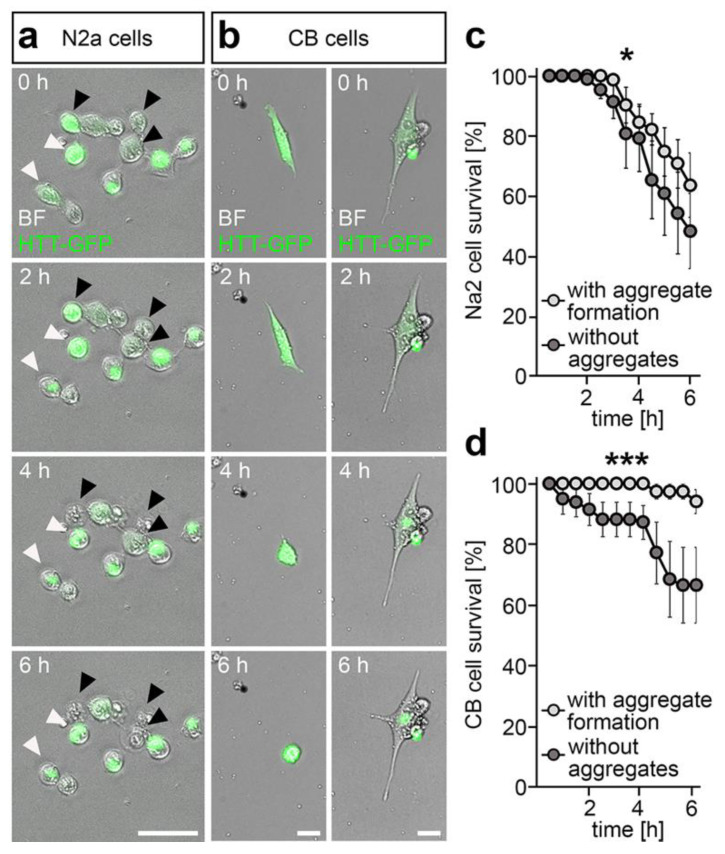
Aggregate formation of mutant huntingtin (HTT) was found to be linked to a higher rate of cell survivability. (**a**) Representative microphotographs of N2a cells expressing GFP-labeled mutant HTT (green) illustrating improved survival for cells with aggregate formation (white arrows) than in cells without aggregates (black arrows). (**b**) Representative microphotographs of cerebellar granule (CB) cells expressing GFP-labeled mutant HTT (green) showing the survival of a cell with aggregate formation (right column), while the cell without HTT aggregate formation was rounded up, indicative of cell death (left column). (**c**) Survival rates of N2a cells expressing GFP-labeled mutant HTT over the time course of 6 h in the HTT cytotoxicity assay discriminating between cells with aggregate formation (*n* = 4 experiments, *n* = 113 cells) and cells without aggregates (*n* = 4 experiments, *n* = 64 cells; two-way ANOVA, * *p* < 0.05). (**d**) Survival rates of CB cells expressing GFP-labeled mutant HTT over the time course of 6 h in the HTT cytotoxicity assay discriminating between cells with aggregate formation (*n* = 27 cells) and cells without aggregates (*n* = 44 cells; two-way ANOVA, *** *p* < 0.001). BF = brightfield, CB = cerebellar granule cells, HTT = Huntingtin, N2a = neuroblastoma cells. Scale bars: 50 µm in (**a**), 20 µm in (**c**).

**Figure 3 ijms-21-05420-f003:**
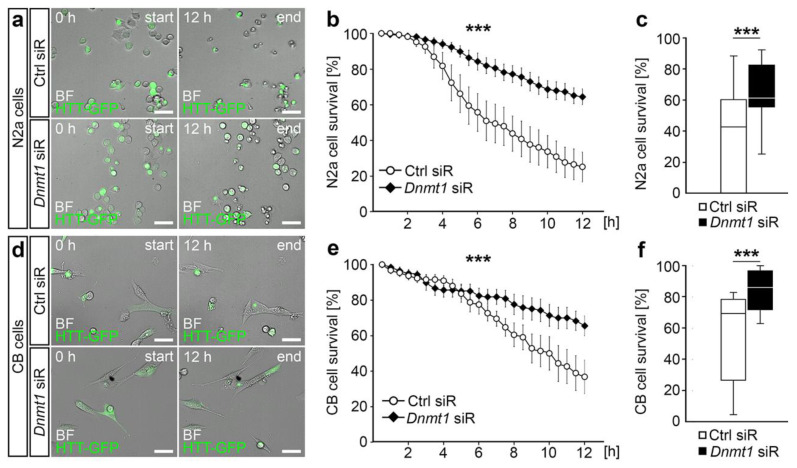
*Dnmt1* knockdown ameliorated the cell survival upon mutant HTT-induced cytotoxicity. (**a**) Representative microphotographs of start and end points of the 12 h time interval used to perform the HTT cytotoxicity assay in N2a cells expressing GFP-labeled mutant HTT (green). Cells were transfected with either Ctrl or *Dnmt1* siRNA. (**b**) Cell survival rate quantification of N2a cells expressing GFP-labeled mutant HTT and transfected with Ctrl siRNA (*n* = 4 experiments, *n* = 274 cells) or *Dnmt1* siRNA (*n* = 4 experiments, *n* = 378 cells) over the time course of 12 h in the HTT cytotoxicity assay (two-way ANOVA, *** *p* < 0.001). (**c**) Comparison of the survival rates of cells analyzed in (**b**) at the final 12 h timepoint (two-tailed Student’s *t*-test, *** *p* < 0.001). (**d**) Representative microphotographs of start and end points of 12 h interval used to perform the HTT cytotoxicity assay in CB cells expressing GFP-labeled mutant HTT (green). Cells were transfected with either Ctrl or *Dnmt1* siRNA. (**e**) Cell survival rates of CB cells expressing GFP-labeled mutant HTT and transfected with Ctrl siRNA (*n* = 100 cells) or *Dnmt1* siRNA (*n* = 132 cells) over the time course of 12 h in the HTT cytotoxicity assay (two-way ANOVA, *** *p* < 0.001). (**f**) Comparison of the survival rates of cells analyzed in (**e**) at the 12 h timepoint (two-tailed Student’s *t*-test, *** *p* < 0.001). N2a = neuroblastoma cells, CB = cerebellar granule cells, Ctrl = control, HTT = Huntingtin, siR = siRNA, BF = brightfield. Scale bars: 50 µm in (**a**,**d**).

**Figure 4 ijms-21-05420-f004:**
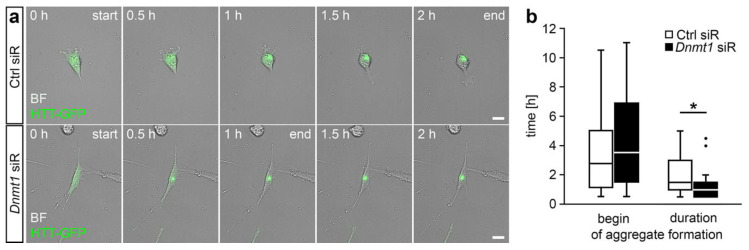
*Dnmt1* knockdown led to a faster aggregate formation of mutant HTT in CB cells. (**a**) Representative microphotographs of CB cells expressing GFP-labeled mutant HTT (green) and transfected with either Ctrl or *Dnmt1* siRNA, showing the temporal differences in the aggregate formation of mutant HTT. (**b**) Quantification of the start and the duration of aggregate formation depicted in (**a**) for cells transfected with Ctrl siRNA (*n* = 27 cells) or *Dnmt1* siRNA (*n* = 33 cells; two-sided Student’s *t*-test, * *p* < 0.05). BF = bright field, Ctrl = control, HTT = Huntingtin, siR = siRNA. Scale bars: 20 µm in (**a**).

**Figure 5 ijms-21-05420-f005:**
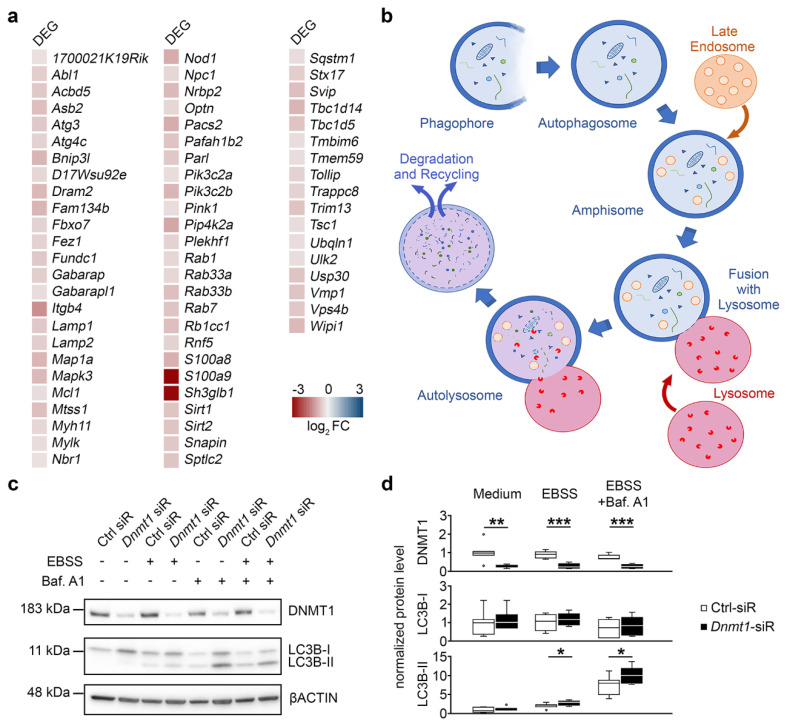
DNMT1 acts on autophagy. (**a**) Heat-map of significantly upregulated genes associated with the GO term *autophagy* in FAC-sorted cortical interneurons of *Pvalb*-*Cre*/*tdTomato*/*Dnmt1* KO mice compared to cells of *Pvalb*-*Cre*/*tdTomato*/*Dnmt1* WT mice. Expression level determined by RNA-sequencing (* *p* < 0.05, Benjamini-adjusted, *n* = 9 WT and *n* = 12 KO mice). (**b**) Schematic illustration of the autophagy pathway beginning with the formation of an autophagosome and depicting the transition to an amphisome, which fuses with a lysosome to form the autolysosome, where particles are degraded. (**c**) Protein levels of DNMT1, microtubule-associated protein 1 light chain 3B-I (LC3B-I), and LC3B-II in Ctrl and *Dnmt1* siRNA-treated N2a cells in the autophagy assay analyzed by Western blot. Cells were cultured in culture medium or starving solution (Earle’s balanced salt solution (EBSS)) and partly treated with an autophagy inhibitor (bafilomycin A1). (**d**) Quantification of protein levels of the conditions shown in (**c**) normalized to beta-actin (bACTIN) and in relation to the control (*n* = 7 experiments; two-tailed Student’s *t*-test, * *p* < 0.05, ** *p* < 0.01, *** *p* < 0.001). DEG = differentially expressed genes, FC = fold change, bACTIN = beta-actin, Baf. A1 = bafilomycin A1, Ctrl = control, EBSS = Earle’s balanced salt solution, LC3B-I/II = microtubule-associated protein 1 light chain 3B-I/II, N2a = neuroblastoma cells, siR = siRNA.

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
