# Peer review of "DNA Methyltransferase 1 (DNMT1) Acts on Neurodegeneration by Modulating Proteostasis-Relevant Intracellular Processes"

_ijms, 2020, doi:10.3390/ijms21155420_

Round 1

Reviewer 1 Report

the paper by Bayer et al. is very interesting and the experimental data have been performed and described adequately.  The only concern is on the description of the methodology about  detection of retrograde vescicle transport that should be better described expecially for the type of microscopy used.

Author Response

We that the reviewer for the overall positive response and have now in more detail explained the methodology of retrograde transportation and microscope settings, as wished by the reviewer. All changes are indicated in red in the manuscript.

Reviewer 2 Report

I have read a manuscript of Bayer et al with great interest. The authors strongly argue that the activity of DNA methyltransferase 1 negatively affects processes of neurodegeneration through the slowdown of retrograde trafficking and autophagy, both being involved in the clearance of aggregation-prone proteins. Importantly, that a knockdown of Dnmt1 ameliorates the neuronal cell survival upon mutant HTT-induced cytotoxicity through the mechanism described above because the HTT protein escapes proteasome-dependent degradation. Thus, I found that the manuscript is of high scientific interest for researchers working in epigenetics, cellular quality control, and neurodegeneration fields.

I was positively surprised by the fact that I don't see any serious points to add from my side. Therefore, I recommend to publish the manuscript in the present form.

Author Response

We thank the reviewer for its positive feedback. As now changes were wished, we have nothing to else to answer here.